# A Survey on UAV Computing Platforms: A Hardware Reliability Perspective

**DOI:** 10.3390/s22166286

**Published:** 2022-08-21

**Authors:** Foisal Ahmed, Maksim Jenihhin

**Affiliations:** Department of Computer Systems, Tallinn University of Technology, 12618 Tallinn, Estonia

**Keywords:** unmanned aerial vehicles, computing platforms, fault analysis, failure modes, cross-layer reliability, fault-resilience

## Abstract

This study describes the Computing Platforms (CPs) and the hardware reliability issues of Unmanned Aerial Vehicles (UAVs), or drones, which recently attracted significant attention in mission and safety-critical applications demanding a failure-free operation. While the rapid development of the UAV technologies was recently reviewed by survey reports focusing on the architecture, cost, energy efficiency, communication, and civil application aspects, the computing platforms’ reliability perspective was overlooked. Moreover, due to the rising complexity and diversity of today’s UAV CPs, their reliability is becoming a prominent issue demanding up-to-date solutions tailored to the UAV specifics. The objective of this work is to address this gap, focusing on the hardware reliability aspect. This research studies the UAV CPs deployed for representative applications, specific fault and failure modes, and existing approaches for reliability assessment and enhancement in CPs for failure-free UAV operation. This study indicates how faults and failures occur in the various system layers of UAVs and analyzes open challenges. We advocate a concept of a cross-layer reliability model tailored to UAVs’ onboard intelligence and identify directions for future research in this area.

## 1. Introduction

Due to the recent unprecedented advances in *Unmanned Aerial Vehicles (UAVs) or drones*, their application has become widespread in public and industrial sectors. Now, drones are used in many areas such as the deployment of wireless networks, product shipping and delivery, precision agriculture, object detection and tracking, border surveillance and monitoring, remote sensing and environmental monitoring, traffic control, and earth mapping [1,2,3]. For instance, recent business insider news reported that the UAV service market size was expected to rise from $4.4 billion in 2018 to $63.6 billion by 2025 and consumer UAV shipments to 29 million in 2021 [4].

At present, the UAV technology is prevalent in many *mission- and safety-critical applications*. E.g., in Search and Rescue (SAR) operations of aftermath disasters, UAVs are employed to seek people who fall in distress or imminent danger [5]. Emergency public safety operations often need the deployment of wireless networks by multiple UAVs at a swarm level. Utilizing UAVs, it is now possible to release humans in inspection and maintenance of dangerous works in the industry, such as power grids, high-power boilers, mines [6]. For these types of mission- and safety-critical applications, multiple UAVs at a swarm level require communication between drones by establishing a wireless network that enables collaborative computing, e.g., by computing tasks offloading [7].

The *reliability* of these mission- and safety-critical applications is inherently connected to the correct service of the UAV system, which consists of several Functional Modules (FMs) such as Flight Control Computer (FCC), Communication Module (COM), Global Positioning System (GPS) module, and different Computation Intensive Payload (CIP) modules for machine learning (ML) methods e.g., Neural Network Accelerator (NNA). These FMs are directly controlled and governed by the *Computing Platform (CP)* onboard the UAV. On the hardware (HW) side, the CP of a UAV can be built of a Micro-controller (μC), Field Programmable Gate Arrays (FPGAs), Microprocessor (μP), application-specific Commercial-Off-The-Shelf (COTS) electronic components, etc.

Failures of UAVs such as position and altitude, crashes with obstacles, and target identification, may happen during the UAV operations due to the soft and hard errors in the CP as well as faults in other parts of the UAV system, such as sensors, actuators, motors employed by the FMs. The consequences may be even catastrophic if failures of UAVs occur in mission- and safety-critical applications. In the scope of this work, we advocate the paradigm of the cross-layer approach of reliability assessment and enhancement discussed in [8,9]. *Cross-layer reliability (CLR)* of a cyber-physical system implies a holistic approach to modeling, detecting, isolation, and recovery of faults originating at each layer and propagating through the other layers of a system. Here, depending on a particular implementation, the system layers may involve the underlying computing HW (processors, accelerators) and their components, the embedded software (SW) and the operating system (OS), the complete single device, and the System of Systems (SoS) performing the application.

Careful consideration of the reliability attribute is essential in designing the complex *fault-resilient* CP of UAVs for *failure-free* operation in mission- and safety-critical applications. To tackle the challenges, cross-layer fault-resilience is currently becoming a potential solution for such a computing system [9]. Most of the UAV survey articles cover various FMs, such as FCC, CIP, and the communication part [6,10,11], and others focus on the different CPs employed in several applications [12,13]. While the reliability problem is currently starting to pose a significant issue, it is only briefly mentioned in a small number of survey works [14,15]. Only in the review [15], the CP and reliability issues were reported; however, the authors limited their discussion to COTS and overlooked the complete system’s reliability. In this survey paper, we fill this gap by studying recent research on both CP used in various FMs and the reliability issues in a cross-layer manner of the UAV system. Furthermore, we highlight the reliability challenges and fault-resilience techniques for failure-free UAV operations. The main contributions of this paper are as follows:We present an overview and analysis of state-of-the-art computing platforms for UAVs;We analyze the reliability challenges and recent fault-resilience techniques for failure-free UAV operation;We outline the concept of the cross-layer reliability model for UAV computing platforms.

The overall structure of this survey work is depicted in Figure 1. At the beginning of this paper, we mention the related work in Section 2 and explore the CPs used in several FMs of UAVs, such as FCC, object detection and tracking, image processing with NNA, and COM in Section 3. Then we discuss reliability assessment studies in Section 4, where we look at various fault and failure modes in UAV systems. Section 5 is dedicated to a review of existing reliability-enhancement strategies. This section also outlines important challenges for developing fault-resilient UAV systems. We present a CLR model for a UAV system to address the reliability challenges and further research direction in this arena in Section 6. In Section 7, we draw the final conclusions.

## 2. Related Works

In this section, we introduce recent survey papers on UAVs in terms of various applications, computing platforms, and related challenges. Then we show the analysis of these reviews and the research gaps that require further attention.

The *security, privacy, and safety* aspects of civilian drones are investigated in paper [6] in which the authors have measured the vulnerabilities of various security-related attacks such as the insertion of malicious activities, and the crashing of the drone. They have analyzed the security requirements of the drone and surveyed existing works in which they offered solutions to such vulnerabilities. However, their analysis is only limited to security-related issues in the cases when attackers send malicious information to the flight control or ground control system via the data link to take control over the drone. Interestingly, security and reliability as extra functional aspects of a complex computing system may have mutual dependencies as studied in survey [16]. While security is a critical issue for UAV systems, it is out of the explicit focus of this survey.

In another survey paper [10], the authors comprehensively reviewed UAVs in many *civil applications* and highlighted the challenges such as charging, collision avoidance, swarming, and security-related in the various subsystems of the UAVs. They have discussed recent technologies such as cloud computing, ML, wireless communication, and image processing used in many UAV applications, for instance, rescue, remote-sensing, precision agriculture, monitoring, surveillance, wireless coverage, etc. Similar to [6], that survey does not take into account the reliability issues in UAV applications. However, both *security and reliability* problems are considered in [14]. The authors have highlighted several challenges and solutions in Neural Network (NN) based Artificial Intelligence (AI) systems, such as energy efficiency, security, and reliability. They focused only on the image processing use cases and did not consider other applications for the UAVs. However, none of the above three survey papers [6,10,14] focused on the CP of UAVs in their discussion.

In a survey work [15], the publicly available open-source *CP of UAVs* and simulators are mentioned including features of functionality, reliability, fault tolerance, and endurance. Although the paper [15] mostly presented the fault-tolerant open-source CP based on μC unit, for several UAVs FMs such as COM, sensors, and actuators, they did not consider the issues of reliability across the layers. The usage of FPGAs-based CP in a UAV is demonstrated in the paper [12]. The authors presented the use of FPGAs in various UAV applications such as navigation, object tracking, and critical mission tasks. They included research works that utilized COTS and FPGAs as CP in various sub-modules of UAVs such as flight control, main controller, communication subsystem, and various payloads. That work advocates the use of FPGAs to perform a variety of computing-intensive tasks, such as object detection and tracking, obstacle avoidance, and so on. However, the authors did not focus on any reliability in their survey analysis.

In papers [11,13], the authors introduced Deep Learning (DL) with Convolutional Neural Networks (CNNs) and other ML methods as the *CIP* in several UAV applications. They discussed the CNN in the ML context and algorithms used in many applications for UAVs such as feature extraction, planning, and motion control. Ref. [11] presented the DL algorithms that consider how to avoid collisions of autonomous UAVs and also presented several DL architectural platforms. The authors also included DL-based operations of the UAV subsystems such as propeller, control system, sensing, positioning, communications, power, storage, and identification. Although the surveys [11,13] studied the timely topic of CIP for several UAV subsystems and applications, they did not consider the reliability issues. Table 1 summarizes recent related surveys on UAV applications and demonstrates the novelty of our survey in this work (the last row in the table) compared to the state-of-the-art.

## 3. UAV Computing Platforms

In this section, we discuss the CP used for the FMs of UAVs in several applications. In general, UAVs can be categorized by using construction and altitude. Depending on the construction, UAVs are either fixed-wing or rotary-wing. While most commercial UAVs are rotary-wing types, fixed-wing UAVs are used for very high-speed operations and can carry much heavier payloads. On the other hand, rotary-wing UAVs can fly at low speeds and has outstanding mobility. The latter type of UAV became popular for many potential applications. In terms of altitude, UAVs also can be found as high-altitude platforms which are deployed for long-endurance surveillance and can fly at altitudes above 17 km and remain almost stationary. Low-altitude UAVs, on the other hand, are designed to move quickly at an altitude of a couple of meters up to a few kilometers [2].

Generally, a UAV is a part of an Unmanned Aerial System (UAS). The following subsystems are the main parts of a regular UAS.
Ground Control Station:

The ground control station acts as the central control unit of the overall UAS, where all the data (video, command, and telemetry) received from the UAV is analyzed and monitored for further decision-making. For smaller UAV applications, communication over a range of up to several kilometers often uses a remote-control system. Satellite systems may be involved in extreme UAV operations where the ground station is located thousands of kilometers from the UAV work zone [17]. Recently, autonomous UAVs or autonomous swarms of UAVs were proposed that may operate in the field without continuous communication with the GCS.
UAV Communication Link:

Data-link or COM is a part of UAVs that provides duplex communication with the ground control station and other UAVs. To safely and reliably operate the UAV, a stable communication system is an important requirement. The COM is mainly composed of a transmitter, receiver, antenna, modulator, etc. Recently the fifth-generation (5G) communication, both the 5G base station-based and device-to-device (side-link) protocols, becomes widely used for this purpose.
UAV Sensor and Actuator:

UAV sensors can be broadly categorized as critical sensors for Inertial Measurement Units (IMUs) and navigation and tracking sensors for route planning and object detection [12]. The critical sensors such as accelerometer, gyroscope, magnetometer, compass, ultrasound height, and pressure sensors are mainly used for flight control to measure the altitude and rotational axis. Image sensors such as video cameras (monocular or stereo), Light Detection and Ranging (Lidar), Radio Detection and Ranging (Radar), and lasers are employed to capture videos and images in path planning and object (stationary or moving) detection. Motors and associated electronics drive circuits act as actuators in UAVs.
UAV Computing Platforms:

Similar to other embedded systems, a UAV needs a CP as a processing system that retrieves data from payloads and other sub-modules. The processed information is then delivered to the actuator and ground station or another UAV to operate the UAV successfully. Most of the CPs of commercial and civil UAVs are μC- or COTS-based embedded systems. However, modern UAVs are performing complex image processing and real-time object detection with the help of ML, DL, and other types of mathematical algorithms. UAVs deployed for applications implying computation-intensive processing use high-speed multi- and many-core processor systems, Graphical Processing Units (GPUs), FPGAs, All Programmable Systems-on-Chip (APSoCs), SoC-FPGA, to process their complex tasks efficiently.

In the following subsection, we will explore CP used for FMs of UAVs in high-level and low-level applications.

### 3.1. Flight Control Computer

The Flight Control Computer (FCC) of FMs plays a significant role in keeping the UAVs in a specific position and returning to the base station properly. If the FCC does not function accurately, there may be a chance of an accident or failing the mission. FCC monitors UAV states continuously through various critical and navigation sensors. The FCC can be sorted by low- and high-level flight control operation [12]. In low-level, basic flight control operations, such as motor control, UAV stability, and processing sensor data are performed. FCC is often engaged in high-level applications such as autonomous navigation, path planning, stereo vision, simultaneous localization, and mapping that make UAVs autonomous. In high-level operations, FCC requires high processing power where an OS is running over HW/SW co-design to implement complex navigation and object detection algorithms. The modern CP such as SoC-FPGA itself can perform both high and low-level operations we denoted them in this paper as hybrid-level, although it also requires the help of other computing devices in critical applications.

#### 3.1.1. Low-Level FCC

*SoC-FPGA:* Low-level FCC FMs such as IMU core, receiver IP cores for pulse-width modulation signal, and Proportional Integral Derivative (PID) controllers were designed and developed on a single SoC-FPGA-based CP in work [18]. In work [19], the authors proposed four techniques in designing the controller of the FCC considering low power, fast response, and less volume for FPGA- or Digital Signal Processor (DSP)-based small UAVs. Research work in [20], presented a secured operation for FCC FM by using μC and FPGA combinedly. μC controls all sensors and generates the signals for controlling the UAV motors. FPGA handles the data encryption and decryption task before sending data to the UAV’s motor and radio systems.

*μC-based FCC:* Using several low-cost sensors such as an IMUs and a Lidar, the research work [21] implemented an μC-based FCC FM for the small rotary-wing UAVs to estimate the position of the UAV and its distance from an obstacle or a landing field. Employing several low-cost sensors such as a 10-DOF micro-electro-mechanical system IMU and a Lidar, research work [22] applied the μC-based FCC FM for small rotary-wing UAVs to determine the location of the UAV and its distance from an obstacle or the landing surface.

#### 3.1.2. High-Level FCC

*SoC-FPGA for Algorithm Implementation:* The authors in work [23] presented SW/HW co-design framework for UAV returning by proposing an improved region-based Kanade-Lucas-Tomasi tracking algorithm. They also improved the hardware acceleration architecture by integrating parallelism and improving resource utilization for FCC FM in the SoC-FPGA-based CP. In the study [24], the authors developed real-time processing systems such as mean subtraction, windowing, finite impulse response filtering, decimation, and spectral estimation via Fast Fourier Transform (FFT). Their implementation results using similar SoC-FPGA CP achieved real-time 3-dimensional detection of local UAV traffic at a range of 1000 m. Similar work is presented by [25] where additional processing system for frequency modulated continuous wave phased array Radar utilizing SoC-FPGA for autonomous navigation to identify nearby aircraft such as small UAVs up to 350 m and bigger aircraft up to 800 m. On that CP, DSP algorithms were also employed, including parallel FFT, cross-correlation, and beam-forming. In work [26], the CORDIC, EKF, and PID-Fuzzy algorithms were integrated with the FCC platform to create a real-time Guidance, Navigation, and Contro (GNC) system on an FPGA to read data from IMU sensors. After processing the payload data, FPGA-based CP generates navigation commands as Pulse width Modulation to actuator and servo motors.

*μC-based High-level FCC:* A decision-making algorithm based on fuzzy logic was demonstrated in [27] using the Arduino Uno μC CP for controlling the IMU of autonomous UAVs. They used an IMU algorithm to predict the parameters of inclination, lateral, and bending angles in flight, which allows the UAV to navigate fast and avoid obstacles. Another study [28] employed an μC-based CP with an embedded flight map containing flight information and constraints on the cargo carried and the flying mode.

#### 3.1.3. Hybrid-Level FCC

*SoC-FPGA-based Hybrid-level FCC:* A model-based HW/SW co-design was proposed in [29] for implementing both high and low-level FCC FMs, where they represented and compared four possible boards to implement such operations. In noisy environments, such as where it is cloudy or under trees, the GPS signals are so weak that UAV faces difficulty in tracking and localization. To tackle these real-time challenges, the work in [30] implemented a real-time vision-based navigation system based on the AprilTag algorithm using the SoC-FPGA CP to perform real-time pose estimation, tracking, and localization in GPS-denied environments. In another similar work [31], the authors presented an approximate adder design focused on error-tolerant size, weight, and power for intensive UAV imaging applications such as 2-dimensional Discrete Cosine Transform, airborne self-localization, and moving object tracking algorithms.

*μC-based Hybrid-level FCC:* The basic FCC operations such as dynamic modeling, control system design, model-in-the-loop, and hardware-in-the-loop of an unmanned helicopter were implemented using a novel Linux-based flight control system built on Raspberry Pi board in work [32]. In [33], the authors focused on implementing an autonomous source-seeking application using Deep Reinforcement Learning on μC-based CP for nano quadcopters. They tested their proposed method using open-source CrazyFile nano quadcopters and found it to be 70% more efficient in source seeking. Using a similar μC-based CP and open-source CrazyFile nano quadcopters, the work in [34] provided an onboard HW/SW autonomous visual navigation system utilizing a CNN-based DL network.

Table 2 shows the list of research works and their implemented FCC FM along with applications. From this table, we can observe that SoC-FPGA platforms are utilized in most of the cases for conducting both high- and low-level operations.

### 3.2. Computation Intensive Payload

In this section, we discuss representative CIP FMs of UAVs such as object detection, tracking, image processing, and NNA applications.

#### 3.2.1. Object Detection, Tracking, and Environment Monitoring

*SoC-FPGA-based Detection:* In work [35], an infrared image processing system was implemented using combined computing platforms of FPGA and DSP for image acquisition, tracking, and matching algorithms. *Terrain classification* is important for an emergency landing, aerial mapping, decision making, and cooperation between UAVs in autonomous navigation systems. Using three algorithms (Gray-Level Co-Occurrence Matrix, Gray-Level Run Length Matrix, and Flow), the research [36] provided a complete solution for terrain classification in differentiating among the four terrain types (water, vegetation, asphalt, and sand). Their proposed solution developed on the FPGA achieved a 95.14% success rate in train classification using the OpenCV library. Another challenge of UAVs in the SAR operation is the *moving target detection*. The authors of [37,38] included speed estimates and object segmentation algorithms to identify real-time moving objects using an area-based image registration method in the SoC-FPGA-based CP.

*μC-based Detection:* In work [39], a moving target detection system was implemented while considering avoiding obstacles robustly in heterogeneous swarm of UAVs. Employing μC-based hybrid controllers, they implemented target seeking and obstacle avoidance calculations separately in a distributed UAV swarm architecture. Similar research work in [40] presented resource-limited platforms using μC and GPU for AI-based object detection and tracking. A CNN algorithm is incorporated where an object tracking algorithm is tailored based on a Gain-Scheduled PID controller to follow the detected object under variable speed.

Sometimes, UAVs are used in safety operations to monitor the surroundings as *environment monitoring*. For instance, authors in work [41] mounted the toxic gas detection sensor array on the IoT-based UAV architecture to monitor the air quality in the given environment. They used μC-based controllers to connect the air sensors and to monitor the sensor data. In a rescue operation using a UAV, people or face recognition is another challenging task that requires a real-time complex processing system. A face detection and recognition system utilizing μC-based CP can identify disastrous people on the ground with high accuracy. In research work [42], authors used the Haar cascade classifier algorithm with OpenCV library in their model and reported that they achieved a 98% True Positive rate for 1.5 m height using the Haar cascade classifier algorithm with OpenCV library in the design. Similar work in [43], the authors used a CNN algorithm for the classification and obtained 100% accuracy with a distance of object 1–4 m in detecting victims of natural disasters. Table 3 shows the different CPs used in various object detection and tracking applications of UAVs.

#### 3.2.2. Neural Network Accelerator

In this subsection, we discuss several representative CPs used as an NNA FM. NNA is, actually, a special processor designed for an artificial NN-based ML workload.

*FPGA as an NNA:* In *computer vision* tasks such as image classification or segmentation, video analysis, and CNN-related DL algorithms are used intensively in many applications. However, the CNN model is challenging to implement in a resource-constrained UAV due to model complexity and costly computing procedures. Many researchers are now employing FPGA-based hardware accelerators to tackle this issue efficiently [44,45,46,47]. In the research work in [44], the author proposed a scalable FPGA-based CNN hardware accelerator for embedded systems based on an 8-bit fixed-point approximation of a hardware-friendly CNN model with the OpenCL framework and obtained 1.9× energy efficiency compared to previous work. Similar works [45,46] described FPGA-based hardware accelerators for implementing depthwise CNN. These research works also achieved better performance than CPU and GPU in object detection. The authors used coarse-grained and fine-grained parallel computing optimization methodologies to improve computational speed and throughput in an FPGA-based CNN accelerator.

A *multi-sensory* fusion technique using infrared and visible light based on CNN for UAV surveillance operations was presented in work [48]. In this study, they built an image fusion approach on two widely used HW accelerators: Zedboard (ARM + FPGA) and NVIDIA TX1 (ARM + GPU), and evaluated the performance, finding that FPGA-based platforms outperform GPU-based platforms. An automated navigation system utilizing both IMU sensors and image processing was employed to estimate the UAV location discussed in the work [49]. They developed a hybrid computing architecture consisting of FPGA, CPU, and μC for carrying out the implementation and data fusion process. In work [50], another multi-sensory fusion task was demonstrated in an energy-efficient way using the Spiking NN on the FPGA-based platform. Their proposed hardware implementation achieved an accuracy of 99.7%.

The research in [51] achieved higher performances using Zynq FPGA over the conventional GPU as an accelerator to implement CNN-based image processing for *real-time object detection* scenarios. To address the issue of the Quality of Experience (QoE), the authors developed an FPGA-based architecture called SCYLLA [52]. SCYLLA offers a novel reconfiguration-based profile generation technique that generates a pool of FPGA design and Deep Neural Network (DNN) model profiles with different QoE performances. They reported that SCYLLA reduces the processing latency by 11.9× and saves 71.5× of the energy consumption compared to the CPU-based solution. Recently, the You-Only-Look-Once (YOLO) method, a fast and accurate DNN architecture, explored new concepts in real-time multi-object recognition. The authors of [53] investigated the performance of several SoC-FPGA platforms in real-time object detection and recognition on the YOLO network. A Tiny YOLOv2 was designed in [54] for the real-time object detection for CNN-based implementation using FPGAs where they achieved 3.19× better than the GPU for the performance-power efficiency. Similarly, in [55], a YOLOv2 NNA was developed on the FPGA platform by designing an accelerator memory access module. Their evaluation proved that the implemented design performs better balance speed and accuracy compared with similar research results.

Table 4 shows several representative FPGA-based CPs used as accelerators in NNA applications for the UAVs.

### 3.3. Communication Module

*SoC-FPGA-based COM:* The authors in work [56], presented an FPGA-based Channel Emulator for Non-Stationary Multiple Input Multiple Output (MIMO) Fading Channels required for UAV *communication system*. They developed several COMs such as a delay module, fading generation, an interpolator for a 2 × 2 MIMO channel implemented in a single FPGA CP which achieved a good performance. A data link terminal controlling several UAVs dynamically was implemented effectively on the FPGA CP in [57] which focused on digital zero-IF signal processing unit design and hardware implementation process. The interleaver module is an important component in the transmitter and receiver module for stable UAV communication. The research work [58] implemented that module on the FPGA CP using LUT RAM. The authors, in work [59], designed an agile digital Software Defined Radio (SDR) system in the SoC-FPGA for the UAV target application. The COMs such as Global Navigation Satellite System, GSM, and WiFi were tested and evaluated on that SoC-FPGA-based CP. In another similar work [60], the authors presented a downlink and uplink high-speed communication in a rapidly changing propagation environment for short-range UAVs. They implemented their proposed design in the SDR system using FPGA and μC.

In [61], the authors proposed a security architecture that uses for UAV *reliable communication* and evaluated COM on the FPGA CP involving the transmission of bitstreams between the UAV and ground station. Similar reliable communication between the UAV and ground control station was built in research work [62] using μC-based CP for a UAV communication system to evaluate the single-carrier Frequency Division Multiplexing (FDM) modulation technique.

An extensive study was performed to investigate the performance enhancement in the UAV-assisted networks for the 5G and beyond 5G wireless communication system [63]. The UAV-assisted networks for 5G wireless communication systems can be a promising solution to deploy emergency wireless communication networks to restore connectivity in post-disaster areas. A model for 5G communication networks was developed for post-disaster wireless networks considering FPGA as the implementation unit of a reconfigurable intelligent reflecting surface to find an optimal power allocation [64,65].

Table 5 represents several CPs used for COMs for the UAVs.

### 3.4. Layers of the UAV Computing Platform

We can conclude that CP is the heart of the UAV system; it controls all sub-systems of the UAV. Based on the above analysis of the CPs used in different UAV applications, we can illustrate a comprehensive layered representation of UAV systems and its subsystems in Figure 2 assuming a swarm intelligent application. The lower layer of the figure represents the UAV edge node, where, in the IoT terms, edge computing is performed considerably reducing time delay and energy consumption when performing a complex task such as real-time object detection [66,67]. The middle layer of this figure represents the UAV swarm intelligence at the fog level, where multiple UAV systems (edge nodes) collaboratively perform real-time complex computing tasks that require offloading technology for the edge UAV to reduce the energy consumption, latency, and throughput [68,69,70]. Efficient communication between multiple UAVs also needs a resource allocation mechanism that can be applied in the UAV networks to maximize the efficiency of the UAV systems [71,72]. The wireless communication networks of UAVs could also be affected by potential cyber-attacks as mentioned [73,74]. Finally, the ground station at the cloud level controls the overall UAV systems by receiving and transmitting the signal. The focus of this survey work is on the CP which is the core processing part for the edge computing of the UAV system as shown in the lower part of the figure. The correct operation of a UAV system is strongly intertwined with the CP’s hardware reliability, necessitating the use of a cross-layer fault-tolerant management system and keeping care of all of the subsystems indicated in the figure. The hardware reliability evaluation for UAVs is discussed in the next section.

## 4. Fault and Failure Modes in UAV System

In this section, we discuss hardware faults in UAV CPs and their effects on UAV systems studied in in recent research works. We have included a total of 12 research papers in our survey for the fault and failure analysis. The papers are listed in Table 6, highlighting fault and the failure modes in different scenarios, CPs, and UAV applications. In our survey work, we categorize the faults shown in the first column of Table 6 based on the paper [75].

In [76,77,78,79,80,81], the authors presented Bayesian Network (BN)-based health management approaches to continuously monitor sensors, software, and hardware components for the detection and diagnosis of UAV failures caused by the environment artifacts. They have analyzed the UAV failures due to GPS and battery usage profiles, HW/SW failures due to the effects of weather disturbance and UAV crashes with birds or other UAVs.

*Actuator and sensor faults* play an important role in bridging control commands and actual control effects. Actuator faults, such as getting stuck, partial loss of effectiveness, and control surface impairments cause the mission failure and collision in the cooperative UAVs, discussed in work [82]. The research in [83], also analyses actuator faults along with a gyroscope sensor fault, using simulation. They reported that the roll, yaw rate, and side-slip angle were significantly affected due to the fault of the sensors and actuator. In [84], three-axis accelerometer faults are described where several failures such as step, ramp, and oscillatory were analyzed in the altitude estimation performance. They also investigated the sensitivity of the attitude estimation performance when varying the error magnitude. In [85], the real-time data from the gyroscope sensor were analyzed where roll rate was monitored to observe the effect of UAV position and altitude due to faulty data.

**Table 6 sensors-22-06286-t006:** Summary of fault and failure modes in UAVs.

Faults	Failure Modes and Effects	Computing Platforms	Sensors/Actuators	Applications	Ref.
Navigation sensors, (Software, Hardware)	UAV accurate position fail, crashes with obstacles	Xilinx FPGA SoC (ZED Board)	GPS, Battery	Critical mission	[76,81]
Actuators gain, bias faults (Hardware)	Degradation of actuator effectiveness, collide with UAVs	–	Actuator	Control multiple UAVs	[82]
Sensors and actuator’s partial loss (Hardware)	Changed the value of roll angle, yaw rate, sideslip angle	ZAGI UAV	Gyroscope, Actuator	Safety mission	[83]
Actuator faults (Hardware)	Altitude estimation failure of step, ramp, and oscillatory error	KARI EAV-3	Accelerometer, IMU	High altitude mission	[84]
Sensors (Hardware)	Affect the stabilization of the UAV altitude and position	Zynq 7000	Gyroscope	Object detection	[85]
Navigation sensors, (Hardware)	UAV altitude and position failure	TopXGun Robotics	GPS, Altimeter, IMU	Navigation	[86]
Soft and Hard Error, Chip (Permanent, Transient)	The vibration of motor, accelerometer become violent, system crash	FPGA, μC	Motor, Accelerometer	SAR mission	[87,88]
SEU (Transient)	Erroneous output, decrease accuracy, classification failure	FPGA	CNN accelerator	Identify and classify the objects	[89]
SEU (Transient)	Image classification error, system crash, vulnerable to operating system	Pynq Z2 FPGA	CNN accelerator	Image classification	[90]
Navigation sensors (Hardware)	Error in angular velocity and acceleration causes high risk of failure	μC	Accelerometer, Gyro, Magnetometer, GPS	Navigation	[91]

The authors of [86] mentioned three types of UAV *navigation sensor faults* (in GPS, IMU), such as point, contextual, and collective that cause the UAV positioning errors. The main reasons for these faults happen actually when the UAV is moving different operational environment. To observe the effect of the error propagation in the inertial navigation system, the work in [91] proposed two models that evaluate overall system reliability, probabilities of particular failures such as accelerators, gyroscope, temperature, and pressure sensors, memory, GPS, etc., which also identify critical components.

To control the accelerometer and motor of the UAV, FPGA is often used as a decision-making controller which is highly susceptible to *transient and permanent faults*, addressed in [87,88]. These faults significantly increase the vibration of the accelerometer and motor of the UAV. FPGA-based CNN accelerator is used in the UAV for object identification and classification task. Radiation-induced soft errors, such as Single Event Upsets (SEU), cause bit flipping in the registers of the implemented CNN accelerators that produce incorrect results and high misclassification rates. In [89], the authors investigated the effects of radiation-induced error on the SRAM-based FPGA where the NN was mounted. Injecting faults such as emulating SEU in the FPGA-based SoC, the effect in CNN was analyzed and found that not all errors need to be considered. Few of them were found tolerable, while others contribute to the overall accuracy drop leading to system failure. In a similar work [90], the authors analyzed the SEU effects by injecting fault and exposing neutron beam on the FPGA-based NN accelerator and identified system crashes, misclassification of images, and vulnerable OS functionalities.

### Analysis of the Fault and Failure Modes of UAVs

Figure 3 depicts the assessment of several UAV faults and failures documented in the previous subsection. The majority of research focuses on faults in several sensors such as altimeters, GPS, and IMU. In many situations, navigation faults in the IMU sensors are also mentioned, along with other sensor types of faults such as GPS and altimeters. Soft errors or transient faults are becoming other sources of many UAV system failures, particularly in the UAV CP, as outlined in Figure 3. In addition to these faults, research studies have looked into other defects in the actuator, motor, and battery subsystems.

In the failure analysis, most of the research works highlighted UAV collisions, position, and altitude mode. Due to soft- and hard-errors, UAVs often fail in target identification problems.

We have highlighted another interesting finding from our survey work depicted also in Figure 3 that the majority of the UAV research utilized SoC-FPGA-based CP for the fault and failure assessment. μC and commercially available UAV platforms are also observed for fault and failure analysis.

## 5. Reliability Enhancement in UAV System

As discussed in the previous section, the CPs control numerous UAV functional sub-modules. An error causing UAV failure may happen at any of the sub-modules due to faults discussed in the previous section. We analysed in total of 29 research papers (including 12 fault and failure analysis papers) in our reliability survey. The papers are listed in Table 7 highlighting several fault-tolerant techniques in different scenarios of the UAV systems. In our survey, the following approaches are found in the recent works for designing the fault-tolerant CP of UAVs in terms of methodology, modeling, and algorithms for reliability enhancement.
Bayesian Networks:

BN are stochastic modeling techniques extensively used to represent and analyze complex systems. In works [77,78], authors presented BN-based health management networks to continuously monitor sensors, software, and hardware components for the detection and diagnosis of UAV failure. Further, they extended their BN-based health management by including an embedded Decision-Making module for UAV mission [76,79]. In another similar work [80,81], the authors incorporated Markov Decision Process with the BN-based model for Failure Mode and Effect Analysis table to evaluate different types of modules. They demonstrated a case study for a target tracking mission that their proposed model can provide Quality of Service (QoS) in missions in hazardous environments.
Markov Chains Model:

Markov Chains Model (MCM) is also another probabilistic model that recently has received much attention in the reliability and safety domain in UAV applications. Ref. [92] provided fault-tolerant models based on MCM for the flight control and navigation system. In addition, they also proposed a reliability synthesis method that allows quickly making rational choices for fault-tolerant systems to meet the required level. Another research work [91] proposed MCM based on designing the flight control system for IMU, in which the authors initially created a system modeling language model and then transformed it into a Dual-Graph Error Propagation Model. Finally, the MCM model was used to evaluate the system dependability matrices. Both soft and hard faults should be taken into consideration when designing fault-tolerant computing architecture, as described in the study [87]. The authors included soft errors such as SEUs and hard faults such as permanent fault models based on MCM and implemented the fault-tolerant re-configurable architecture on the FPGA- and MC-based CP. In addition, Principal Component Analysis (PCA) is used to classify a UAV’s health conditions based on the accelerometer data. Similar research [88] also investigated both soft and hard faults in a more extended way in terms of reliability, power consumption, and system weight when Continuous Time MCM is used to estimate reliability. They included Dynamic Partial Reconfiguration (DPR) for the recovery cases when faced with soft or hard errors and chip failures.
Kalman Filter:

Kalman filter, also known as linear quadratic estimation, is an algorithm that has numerous applications such as guidance and navigation, vehicle control, specifically aircraft, and UAVs. In [84], a robust dynamic model-based estimator was proposed to estimate the states and faults of the three-axis accelerometer using the Kalman filter algorithm. The authors of [86] mentioned a fast and accurate fault detection technique for onboard navigation sensor faults. They employed a Kalman filter to estimate real-time model-free residual analysis and a data-driven Adaptive Neuron Fuzzy Inference System to build a reliable fault detection system. The failures of sensors and actuators for UAVs were investigated in a fault-tolerant flight control system using an adaptive Kalman filter [83]. Their proposed design also can isolate the sensors when found any fault. Ref. [93] achieved optimization at the collaborative position of the faulty UAVs due to GPS faults by employing extended Kalman filters.
Automata:

In [94], the authors presented a novel framework based on statistical model checking with composed Priced Timed Automata for the reliability analysis of UAV. To measure the reliability of UAV-UAV communication, several Automata models are introduced for the communication modules such as UAV’s transmitter, receiver, data exchange, and replacement model. A comparable DPR design strategy for UAVs for reliable autonomous management was reported in work [95] using the Automata model.
Neural Network:

In the previous section, we have seen many applications of the NN model in the image processing for the object detection and path planning task of autonomous UAVs. Now, we will discuss how the NN model can also contribute to making the UAV fault-tolerant. In [96], the authors reported a fault diagnostic system based on a hybrid feature model and DL to monitor the sensor’s fault. In a similar work [85], a DL-based fault diagnostic system was also reported considering real-time fault detection and employing a PCA technique to improve computing efficiency in the DNN implementation. In another work in [97], an improved algorithm was proposed for fault-tolerant IMU considering the reduction of redundant information processing in the NN operation. In [89], the authors investigated the effects of radiation-induced error on the SRAM-based FPGA, where the NN was mounted. They reported that their proposed quantized CNN layer is 39% less sensitive to radiation. A similar quantized CNN method analyzed the impact of SEU on the reliability of the proposed CNN on the FPGA-based CP that includes a Triple Modular Redundancy module in [90].
sensors-22-06286-t007_Table 7Table 7Summary of reliability enhancement in UAV system.ApproachSafety and Reliability EnhancementApplicationSensors/Actuator/ModuleComputing PlatformsRef.Bayesian networkDecision making including failure managementCritical missionGPS, BatteryXilinx SoC-FPGA (ZED Board)[76,80]Decision making failure managementCritical missionGPS, BatteryXilinx FPGA SoC (ZED Board)[78]Decision making failure managementCritical missionGPS, BatteryXilinx FPGA SoC (ZED Board)[79]Embedded health managementCritical mission computingAccelerometerXilinx ZED FPGA[77]Fault detection, isolation, and recoveryCritical missionGPS, BatteryXilinx Zynq FPGA[81]MCMReliability synthesis for flight computerNavigationFCC–[92]Fault-tolerant architectureSAR missionMotor, AccelerometerFPGA, μC[87,88]fault-tolerant inertial navigation systemNavigationIMUμC[91]KalmanRe-configurable fault-tolerant controlSafety missionActuator–[83]Fault-tolerant accelerometerHigh altitude missionAccelerometerKARI EAV-3[84]Fault-tolerant cooperative systemNavigationGPS, Radar, IMU–[93]Sensor and navigation fault detectionNavigationIMU–[86]AutomataResource management for safety purposesVideo trackingCameraXilinx FPGA ARM, Neon processor[95]Statistical framework for SEUUAV communicationCOM–[94]Neural networkFault detection for sensorsNavigationGPS, IMUUltra-Stick 25e UAV simulation model[96]Reliable CNN for FPGACNN acceleratorAcceleratorXilinx Zynq FPGA[89]Decision making failure managementGeneralFCCXilinx Zynq FPGA[85]Analysis SEUGeneralOn-chipPynq Z2 FPGA[90]Fault-tolerant neural networkMissionIMUFPGA, μC[97]Fuzzy logicFault-tolerant quadcopterSAR missionFCCFPGA, μC[98]Tracking algorithmFailure detection and identificationVisual inspectionCameraOdroid U3[99]Polygonal linear consecutiveMission reliabilityMissionNode-based–[100]Cooperative control modelFault-tolerant for cooperative droneControl multiple UAVsActuator–[82]Model-free controlAlgorithmic optimizationControlling in unstructured environmentsUnderactuated manipulator–[101]Event-triggeredResource optimizationNetworked control systemsActuator–[102,103]Unified modelingAutomatic testing platformReal-time fight simulationFCCPixhawk autopilot[104]
Other Techniques:

Here, let us consider other research works that focus on different methods for developing reliable CP for UAVs. Another fault-tolerant cooperative control was designed in work [82] for the safety of multiple UAVs based on sliding mode techniques where they investigated the scenarios involving actuator faults. In the paper [101], the authors addressed the model-free intelligent and adaptive control approaches that can also handle the failures and faults in the presence of various parametric and non-parametric uncertainties. An event-triggered control is another effective control solution that can be used for saving limited computation burden, battery power, and control cost of electrical devices for UAV reliability enhancement as reported in [102,103]. A cooperative virtual sensor was established for vision-based fault detection and identification in multiple UAV applications, as discussed in [99]. Another fault-tolerant FC considering both SEU and total chip failure was developed in work [98]. In addition, an obstacle avoidance system has been established using Fuzzy logic and investigated its performance through MATLAB tools. In work [100], the authors investigated Mission reliability modeling for UAV swarms using the polygonal linear consecutive system that analyzed the performance in terms of different reliability and the structure optimization of UAV swarms. An automated test platform based on a unified modeling method was presented and developed as a real-time simulation platform by employing automatic code generation and an FPGA-based hardware-in-the-loop simulation method [104].

### 5.1. Analysis of Reliability Enhancement

Figure 4 depicts a summary of many reliability enhancement techniques utilized in the development of fault-tolerant UAV systems. For designing fault-tolerant to different problems such as sensor and actuator errors, Bayesian and Markov chain-based models are employed in the majority of instances. Many research employed the Kalman filtering method to construct fault tolerance tracking and guided navigation systems. The DPR approach has also been employed in the in-field resilience enhancement methods in a CNN-based accelerator for reliable object recognition and classification tasks to deal with soft- and hardware error problems.

Figure 4 also shows CP used for the reliability analysis where SoC-FPGA is used in most of the cases, similar to the fault analysis stated in the preceding section. Finally, we may deduce from the foregoing observations of reliability improvement works that it is critical to managing the faults in all layers of the UAV system when constructing a fault-tolerant UAV system.

### 5.2. Challenges in Reliability Enhancement of UAV

It is clearly observed in the previous section that faults or errors may occur at several layers in the UAV system. From Figure 5, we summarize various layers in the UAV system to mimic the possibilities of fault occurrence scenarios based on the recent research analysis described in the previous section. The reliability threats for computing platforms are mainly due to radiation-induced faults such as SEUs (soft errors) and hardware permanent errors, e.g., the ones due to the nanoelectronics aging phenomenon [87,88,89]. Another important threat may happen in the sensors and actuator layers due to the electro-mechanical fault and harsh environment [86,91]. A UAV system may face system failure due to the effect of any of the above two failures or both. For instance, due to any of these faults, the UAV may send the wrong information to the GCS or other UAVs, UAVs unexpectedly fall in the civil area or collide with other objects (UAVs, obstacles, etc.). In the case of multiple UAV swarm intelligence, these scenarios eventually may cause mission failure as SoS failure [10,39]. As a result, we require a UAV system “health” management that continuously monitors and analyzes faults and escaped errors, isolates or mitigates them, and maintains the mission operational continuously even when individual system components fail. We have identified the key challenges in developing reliability enhancement of UAV in the HW perspective:Nanoscale implementation technology:

The current technological shrinking tendency of the devices makes UAV CPs highly error-prone. The new, non-validated technologies and complex architectures imply a variety of possible hardware-originated faults that may manifest themselves in each of the layers shown in Figure 5. Such faults propagate across the system layers and cause masked or non-masked (escaped) errors along the way.
Chip-level hardware architectures:

Another point of concern is UAV hardware architecture that presently includes General-Purpose GPUs, many-core processors, APSoCs, Tensor Processing Units (TPUs), Intelligence Processing Units, and SoC-FPGAs, etc., for the processing of CIP at the UAV edge node discussed in Section 3. This scenario raises several important issues, including HW/SW design complexity, adequate knowledge of their response to possible faults, and effective new fault models. Secondly, detection and possibly tolerating faults should be identified, taking into account that they should be cost-effective and have a shorter time-to-market.
Complex architecture of UAV computing:

The intelligent UAVs deployed for computing-intensive applications imply a highly complex distribution of computing tasks within a single UAV or event collaborative computing hierarchical SOS architectures in the case of UAV swarms, as shown in Figure 2. Currently, there is no established comprehensive reliability modeling methodology supporting the distribution of computing in a holistic manner. Specifically, collaborative computing in a swarm of UAVs needs optimized solutions in constrained resource utilization for real-time applications.
Autonomy requirements of UAV:

UAVs still do not have complete autonomy. Most of them are semi-autonomy levels, in which several UAV flight functions, such as collision avoidance, object detection and tracking, run autonomously using AI. More research is needed to develop the decision-making capabilities of UAVs. Safety- and mission-critical inspections for autonomous UAVs have certain limitations, such as weather, accessibility, weight, and regulations. Similarly, autonomous fault detection capability, in-field failure analysis, adaptive to a fault, and resource allocation are significant attributes in developing an autonomous UAV.
Constrained resources of a mobile device:

Another challenge to the reliability of UAV systems is the accurate assessment and management of the limited resources, such as battery power, the slacks of real-time execution time, etc. for the reliability overheads, i.e., the redundancy for the fault tolerance. The solutions should be capable of efficiently and dynamically managing the system-level performance and priorities in runtime.
Standards:

The International Organisation for Standardisation (ISO), ISO TC20/SC16 [105] specifies several requirements in the field of UAS, including classification, design, manufacture, operation (including maintenance), safety, and traffic management of UAS operations. Many standards are still under development, such as test methods for civil multi-copter UAS, guidelines for UAV testing/design of accelerated life cycle testing for UAS, and test methods for flight stability of multi-copter UAS in challenging environments.

## 6. Cross-Layer Reliability of UAV Computing Platforms

To tackle the challenges of establishing reliability for UAV computing platforms, cross-layer reliability approaches similar to [8,9] is a promising solution.

Figure 6 shows the overall concept of a cross-layer reliability model for the UAV CP. This hierarchical CLR model integrates several heterogeneous self-health awareness parts Figure 6a. We have seen in the previous section that failures may happen at the different layers of the UAV system. The CLR model considers the faults from the lower layers of CP (HW, embedded SW) up to the SoS layer. This approach aims at the ability of the UAV system at each of the layers to comprehend and maintain its health by monitoring the fault-related information, as well as adapting to any dynamic changes when the fault occurs at any layer. This health information is then propagated to the higher layer to adapt the specified operations, thus preventing failures and disastrous consequences.

The CLR model is depicted more elaborately from the chip to the system level than as simple in Figure 6b. Depending on the system complexity, the health monitoring system in the CP architecture can be implemented within a single SoC [76] or expanded over several hierarchical layers in a large system [30,31]. The hierarchical levels in the complex system may require several devices (μP, μC, SoC-FPGA, GPU, TPU, etc.) located on one or more PCBs (see ➀ in Figure 6b). The embedded (on-chip) instruments or monitors access the health data from several sensors, internal and external, using e.g., IJTAG [106] (see ➁ in Figure 6b) and convey the health information to the local controller called Health Management (HM) as an on-chip self-health monitoring system. According to the fault and resource management at various sub-modules, the device can include many embedded instruments or/and HM controllers or system-level controllers.

To improve the resiliency of the UAV edge node against the reliability threats such as aging, soft or hard error, effective mitigation techniques are required. Thus, the local embedded controller reports the health status to the central higher controller in the hierarchy. Fault monitoring and resilience, which includes fault or error detection, analysis, classification, and decision or fault handling, can be carried out in the on-chip or off-chip embedded SW (see ➂ in Figure 6b).

The HM is an SoC [76,107] that contains several on-chip sensors for in-field health monitoring such as aging, temperature, voltage, soft/hard error, and other sub-modules (payloads, sensors, and actuators) of UAVs (see ➃ in Figure 6b). Built-in-Self-Test can also be created by employing embedded instruments connected to the IJTAG network to monitor on-chip health [108]. Health information from UAV sub-modules can be accessed through conventional ports (I2C, CAN, UART, etc.) and an extended IJTAG network [109]. The Decision Manager (DM) in the embedded SW acts as a virtual (processing system) sensor or actuator [110] that makes decisions based on indirect measurements of abstract conditions, contexts, inferences, or estimations from on-chip sensors or external sensors in the UAV sub-modules (see ➄ in Figure 6b). This DM network is a software/hardware co-design that acts as a virtual actuation and can predictably influence system design objectives such as performance, power, and reliability, as well as system QoS [111]. By incorporating current techniques such as Bayesian Network, Markov Model, Kalman Filtering, and CNN, the reliable HM model can be mapped as stated in the previous section [76,89,104,112].

It is worthwhile to consider the performance of the UAV mission and QoS firmly when integrating the aforementioned CLR model into the UAV system. Resource optimization and self-awareness attributes are valuable requirements when designing an energy-efficient reliability model for a specific UAV application.

## 7. Conclusions

With technological development, UAVs can perform many tasks such as object detection and tracking, product delivery, agriculture and environmental monitoring, which demand reliable operation. This paper represents the research results of UAV computing platforms such as SoC FPGA, ASIC, GPU, μC, along with their FMs used in these applications. Both CP and sensors/actuators in the FMs encounter faults that cause UAV failures such as crashes with obstacles, position, altitude, incorrect classification, etc. Defects in one module of a UAV may induce errors in another module. Owing to the heterogeneous system architecture of UAVs, constraints, technology, and different standards, it is challenging to build a fault-tolerant CP when considering the faults holistically.

Recent fault-tolerant techniques focus on either sensor/actuator faults or on-chip defects. Considering the effect of these faults in all sub-modules of UAVs on health monitoring requires careful attention for reliable operation. UAVs must be capable of self-adapting to defects and faults in safety-critical applications. Obtaining CLR and self-awareness for UAV CP by integrating all health information is essential for scientific research in developing fault-tolerant CP for UAVs.

## Figures and Tables

**Figure 1 sensors-22-06286-f001:**
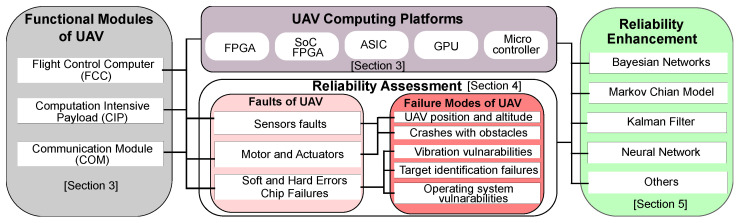
The overall structure of this survey.

**Figure 2 sensors-22-06286-f002:**
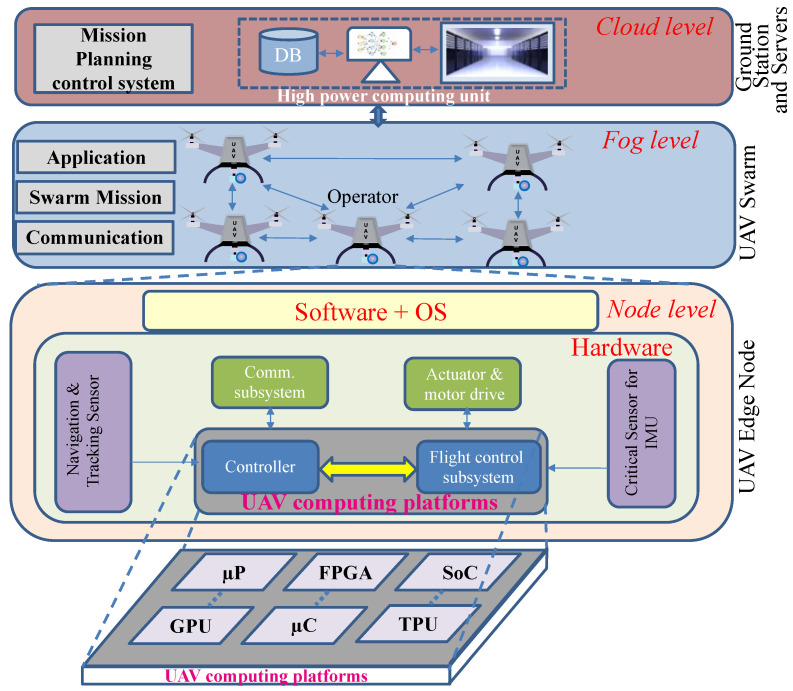
Basic overview of UAV system.

**Figure 3 sensors-22-06286-f003:**
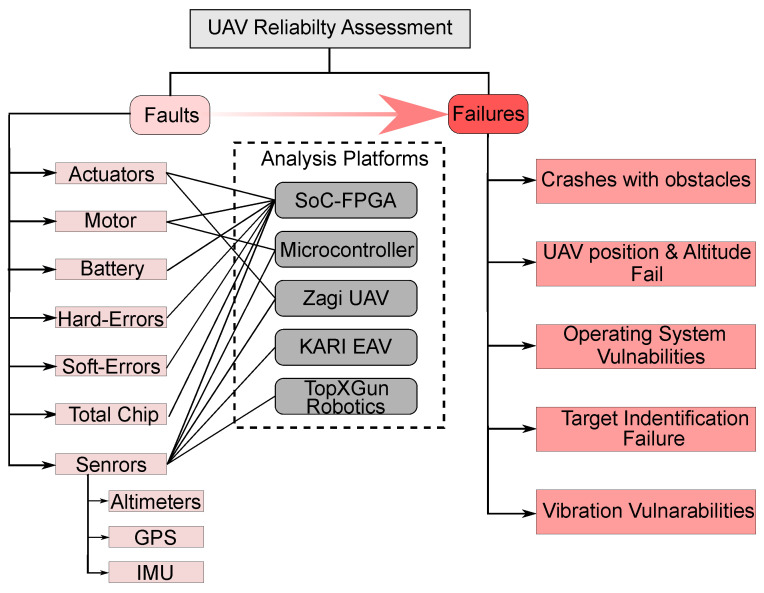
Fault and failure modes analysis taxonomy.

**Figure 4 sensors-22-06286-f004:**
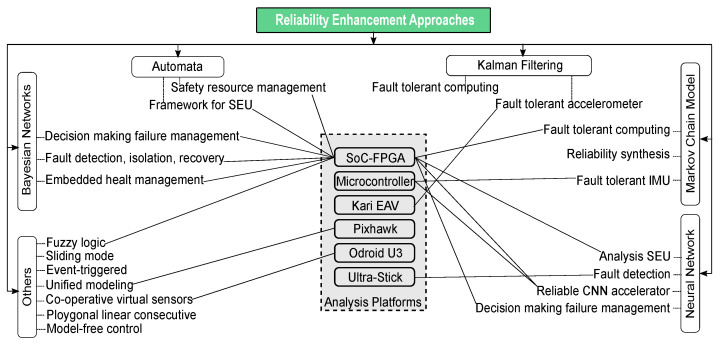
UAV computing platform reliability enhancement taxonomy.

**Figure 5 sensors-22-06286-f005:**
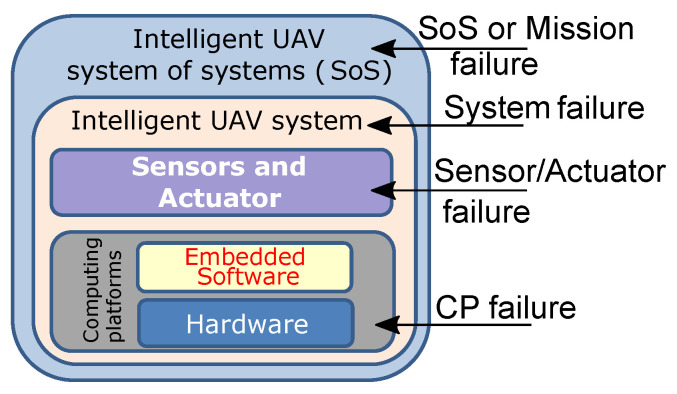
Different system layers and possible failures.

**Figure 6 sensors-22-06286-f006:**
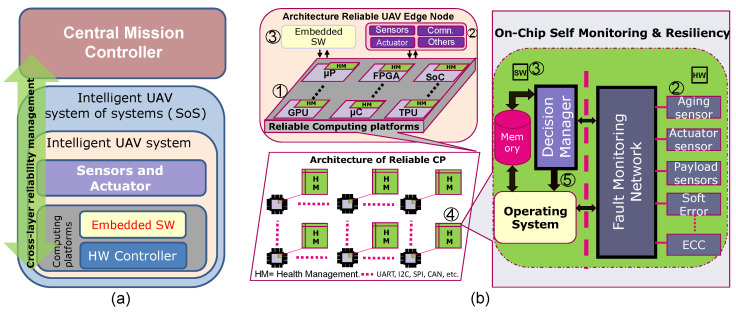
Cross-layer reliability model. (**a**) Cross-layer reliability modeling at different layers of UAV system. (**b**) System architecture of reliable UAV edge node, computing platforms, and on-chip health monitoring system.

**Table 1 sensors-22-06286-t001:** Comparison of recent related works.

Paper Contributions	Functional Modules			Computing Platforms			Dependability			Ref.
	FCC	CIP	COM	FPGA	μC	COTS		Reliability		
									CLR	
Security and safety	✗	✗	✓	✗	✗	✗	✓	✗	✗	[6]
Challenges for civil applications	✗	✓	✓	✗	✗	✗	✗	✗	✗	[10]
Image processing NN and reliability	✗	✓	✗	✗	✗	✗	✓	✓	✓	[14]
COTS and simulator	✓	✗	✓	✗	✓	✓	✓	✓	✗	[15]
Survey of FPGA application	✓	✓	✓	✓	✗	✗	✗	✗	✗	[12]
UAV subsystems	✓	✓	✓	✓	✓	✓	✗	✗	✗	[13]
General purposes and algorithms	✗	✓	✗	✗	✗	✗	✗	✗	✗	[11]
CP and reliability aspect	✓	✓	✓	✓	✓	✓	✓	✓	✓	Prop.

**Table 2 sensors-22-06286-t002:** List of research works and their implemented FCC FMs.

CP	Devices	Sensors/Actuators	Applications	Ref.
FPGA	Xilinx Zynq SoC	C, L, R, IMU	Payload data processing	[18]
		GPS	Navigation	[23]
		R	Flight computation	[24]
		C	Estimation, tracking, localization	[30]
	Xilinx Artix7, Virtex-V, Cyclone II	IMU	Flight control computing	[19]
	Xilinx Virtex-7	C	Moving target detection	[31]
	Xilinx Virtex-7, ZED Board, Raspberry Pi	C, L, R, IMU	Flight and payload computing	[25,29]
	Intel DE0 nano FPGA	C, L, R, IMU	Flight and payload computing	[26]
μC	Arduino Uno	IMU	Flight and navigation	[27,33]
	PIC 32	IMU	Flight and navigation control	[28,29]
	Cortex-M4	IMU	On-board flight control	[34]
		R, L	Flight controller	[21]
	Arduino Mega 2560	L, IMU	Flight and navigation control	[22]
	Arduino Mega	IMU	Flight computing	[20,32]
*R = Radar, L = Lidar, C = Camera*				

**Table 3 sensors-22-06286-t003:** Computing platforms used in object detection and tracking applications.

CP	Devices	Sensors	Applications	Ref.
FPGA	Spartan-3A DSP XC3SD1800A	Camera	Terrain classification	[36]
	Intel i5 CPU, an Nvidia GTX1070 GPU		Target tracking and recognition	[37]
	Cyclone III, TMS320C6657 DSP		Image acquisition	[35]
	Xilinx Ultrascale+ MPSoC		Realtime moving target detection	[38]
μC	ATMEGA 328	Gas detector	Environment monitor	[41]
	ARM Cortex M4, NVIDIA Jetson TX2	Camera	Object detection and tracking	[40]
	Raspberry Pi		Face detection and recognition	[42]
			Disaster people recognition	[43]
			Target detection, obstacles avoidance	[39]

**Table 4 sensors-22-06286-t004:** The summary of FPGA-based NNAs used in UAVs.

CP	Devices	Sensors	Applications	Ref.
SoC-FPGA	Xilinx Spartan-6, Raspberry Pi	IMU, Camera	Autonomous navigation by data fusion	[49]
	Xilinx Pynq-Z1	GPS	Object detection, SAR	[47]
	Xilinx Zedboard, NVIDIA TX1	Infrared and visual	Object detection by Image fusion	[48]
	Xilinx FPGA	Laser, Radar	Cropland monitoring	[50]
	Xilinx ZCU102 FPGA	Camera	Vehicle counting	[52]
	Zynq Ultra Scale		Road object recognition,	[51]
			Vision-based navigation by YOLO	[53]
	Arria 10 FPGA, Intel core I5 CPU		Target detection	[46]
	Intel Cyclone V FPGA		Image classification	[44]
	Xilinx Virtex7 xc7vx690		Object detection, SAR	[45]
	Digilent NetFPGA-SUME FPGA		Object detector by YOLOV2	[54]
	Xilinx KU115		Target detection by YOLOV2	[55]

**Table 5 sensors-22-06286-t005:** Computing platforms used in UAV communication module.

CP	Devices	Communication Technology	COM	Ref.
FPGA	Xilinx Virtex-7	MIMO	Non-stationary channel model	[56]
	Xilinx Artix-7	TDMA	Datalink terminal	[57]
		Spreading, jamming	Variable feedback controller	[61]
	Xilinx Zynq	OFDM, CDM	SDR system	[59]
		OFDM, MIMO	SDR system	[60]
	–	Interleaving	Interleaver module	[58]
	–	5G wireless communication	Intelligent reflecting surface	[64,65]
μC	ArduPilot Mega	Single-carrier FDM, OFDM	Datalink terminal	[62]

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
