# Peer review of "A Survey on UAV Computing Platforms: A Hardware Reliability Perspective"

_sensors, 2022, doi:10.3390/s22166286_

Round 1
Reviewer 1 Report
The paper provides a detailed review of the UAVs. Please consider these points to improve the paper further:
** Please provide more accurate and informative title for the paper.
** English of the paper should be improved as well. There are some obvious grammar mistakes. The articles should be added as well
2 **Abstract of the paper should be improved. The first sentence can state the importance of the content, then the gaps in the corresponding literature. Key contributions of the paper should be expressed clearly and then the major findings of the paper should be provided.
3 **Introduction has provided some background researches and highlighted their advantages and disadvantages. However, critical review of the recent and related works are not quite strong. The corresponding gaps should be emphasized strongly and based on these gaps, the claimed contributions of the paper should be justified.
**Please note that the comparisons of the various algorithms should be performed under equal conditions. Stating the advantages and disadvantages are important, but they should be assessed by considering the conditions that they are performed.
** Please explain what "Cross-layer reliability" implies.
** The statement "Object detection with tracking" is vague.
** The paper reviews the faults and failures which is still a hot research topic. The paper should also address the control approaches that can handle the failures and faults in the presence of various parametric and non-parametric uuncertainties. Model free intelligent and adaptive control approaches can hadle such failures in the presence of the uncertainties. I would suggest this recent and related paper; A novel exploration-exploitation based adaptive law for model free control approaches.
** The paper covers descriptions and reviews in a mixed order which cause some problems for the readers.
** The paper should be re-organized. Som of the main and sub-sections are quite long, and the paragraphs do not folow each other properly. Therefore, understanding and following of the provided information are quite challenging.
Good luck with the improvements...
Reviewer 2 Report
This paper presents a systematic and detailed review, oriented towards UAV computing platforms, besides, it also analyses the specific faults and failure modes on UAV and existing approaches dealing with the reliability aspect. However, there are several problems in the paper, which make the paper cannot be fully appreciated. Improvement is expected if the authors can take the following points into account.
1) Edge cloud computing is playing a more and more important role in the UAV filed, due to the rising complexity and diversity of special applications based on UAVs. It can greatly reduce the time delay and energy consumption in executing complex tasks, which can improve the reliability of UAV systems. So, the author should introduce it in the paper. Edge cloud computing is used in [1] [2] to help UAVs preform complex tasks like object detection and should be quoted in the paper.
2) Computation offloading is an important technology in edge cloud computing, which can largely reduce the energy consumption of UAVs and improve the reliability of UAV systems. [3] proposes a scheme which can effectively improve utilities of UAVs and reduce the average delay. Special optimization algorithms for computation offloading are proposed in [4] [5] for UAV applications. Besides, [6] proposes a learning-aided cooperative offloading mechanism, [7] presents a jointly optimized design of cooperative placement and scheduling framework for computation offloading. These can also be quoted in the paper.
3) Resource allocation is another important technology in edge cloud computing which can be applied in UAV networks for performing large and complex tasks. [8] proposes a dynamic resource allocation method for multipe UAVs enabled communication networks. [9] designs the resource allocation strategy and the trajectory of the UAV which maximizes the system energy efficiency and makes the working system more secure. [10] proposes an adaptive resource allocation approach based on reinforcement learning. The above papers can also be quoted.
[1] Salhaoui M, Guerrero-González A, Arioua M, et al. Smart industrial iot monitoring and control system based on UAV and cloud computing applied to a concrete plant[J]. Sensors, 2019, 19(15): 3316.
[2] Bhoi S K, Jena K K, Panda S K, et al. An Internet of Things assisted Unmanned Aerial Vehicle based artificial intelligence model for rice pest detection[J]. Microprocessors and Microsystems, 2021, 80: 103607.
[3] Chen W, Su Z, Xu Q, et al. VFC-based cooperative UAV computation task offloading for post-disaster rescue[C]//IEEE INFOCOM 2020-IEEE Conference on Computer Communications. IEEE, 2020: 228-236.
[4] Zhang J, Zhou L, Zhou F, et al. Computation-efficient offloading and trajectory scheduling for multi-UAV assisted mobile edge computing[J]. IEEE Transactions on Vehicular Technology, 2019, 69(2): 2114-2125.
[5] Zhan C, Hu H, Sui X, et al. Completion time and energy optimization in the UAV-enabled mobile-edge computing system[J]. IEEE Internet of Things Journal, 2020, 7(8): 7808-7822.
[6] Li Y, Wang X, Gan X, et al. Learning-aided computation offloading for trusted collaborative mobile edge computing[J]. IEEE Transactions on Mobile Computing, 2019, 19(12): 2833-2849.
[7] Li Y, Dai W, Gan X, et al. Cooperative service placement and scheduling in edge clouds: A deadline-driven approach[J]. IEEE Transactions on Mobile Computing, 2021.
[8] Cui J, Liu Y, Nallanathan A. Multi-agent reinforcement learning-based resource allocation for UAV networks[J]. IEEE Transactions on Wireless Communications, 2019, 19(2): 729-743.
[9] Cai Y, Wei Z, Li R, et al. Joint trajectory and resource allocation design for energy-efficient secure UAV communication systems[J]. IEEE Transactions on Communications, 2020, 68(7): 4536-4553.
[10] Yang Z, Nguyen P, Jin H, et al. MIRAS: Model-based reinforcement learning for microservice resource allocation over scientific workflows[C]//2019 IEEE 39th international conference on distributed computing systems (ICDCS). IEEE, 2019: 122-132.
Reviewer 3 Report
This survey article studies the UAV CPs deployed for representative applications, the specific faults and failure modes and existing approaches dealing with the reliability aspect such as fault-resilience mechanisms in CPs for 10 failure-free UAV operation. The investigated topic is interesting and meaningful. However a major revision according to the following detailed comments is needed to improve the quality of the paper.
1. There are two many abbreviations, which are confusing for the readers to follow the paper. It is suggested to add a list of all abbreviations for referring after Abstract.
2. In this paper, only three challenges in developing reliability enhancement of UAV are discussed. As a survey paper, it seems the discussions are insufficient and more challenging problems of developing reliability enhancement of UAV are required to be provided.
3. It is known that the communications between drones are realized by wireless networks, which could be attacked by hackers. Besides the faults of UAVs, this can be viewed as another type of the reliability problem. Some relevant results about this topic are listed for referring: Probability-density-dependent load frequency control of power systems with random delays and cyber-attacks via circuital implementation; Nonfragile integral-based event-triggered control of uncertain cyber-physical systems under cyber‐attacks; Co-design of event-triggered scheme and H∞ output control for Markov jump systems against deception attacks.
4. Recently, event-triggered scheme has been an interesting topic of saving limited computation burden, battery power and control cost of electrical devices. More future research topics regarding the event-triggered control problems of UAVs are suggested to be added. Some recent references are given as: Event-triggered H∞ control of networked control systems with distributed transmission delay; Memory-event-triggered H∞ filtering of unmanned surface vehicles with communication delays; Derivative-based event-triggered control for networked systems with quantization.
5. English presentation needs to be polished. There exist some grammar errors and typos, please check the presentation carefully.
Round 2
Reviewer 1 Report
The paper has been revised and can be accepted.
Reviewer 2 Report
no further comments
Reviewer 3 Report
I have no further comments,it can be accepted now.